# Infected Shoulder Arthroplasty in Patients Younger than 60 Years: Results of a Multicenter Study

**DOI:** 10.3390/microorganisms11112770

**Published:** 2023-11-14

**Authors:** Adrien Jacquot, Ramy Samargandi, Lisa Peduzzi, Daniel Mole, Julien Berhouet

**Affiliations:** 1Centre for Chirurgie des Articulations et du Sport (ARTICS), 24 rue du XXIème Régiment d’Aviation, 54000 Nancy, France; ad.jacquot@gmail.com (A.J.); d.mole@wanadoo.fr (D.M.); 2Service de Chirurgie Orthopédique et Traumatologique, CHRU Trousseau, Faculté de Médecine de Tours, Université de Tours, 1C Avenue de la République, 37170 Chambray-les-Tours, France; ramy.samargandi@hotmail.com; 3Department of Orthopedic Surgery, Faculty of Medicine, University of Jeddah, Jeddah 23218, Saudi Arabia; 4Service de Chirurgie Orthopédique, Centre Hospitalo-Universitaire Nancy-Emile Galle, 49, rue Hermite CS 5211, 54052 Nancy, France; peduzzi.lisa@gmail.com

**Keywords:** shoulder arthroplasty, shoulder prosthesis, prosthetic joint infection, Cutibacterium acnes, infection

## Abstract

**Background:** Periprosthetic joint infection (PJI) after shoulder arthroplasty remains a significant complication. This study aimed to explore the epidemiology and risk factors of shoulder PJI in patients aged 60 and younger, analyze treatment options, and evaluate outcomes after 1-year follow-up. **Methods:** In this retrospective multicentric observational study, data from 1404 shoulders in patients under 60 who underwent primary shoulder arthroplasty were analyzed. Patients with PJI and at least 1-year follow-up after infection treatment were included. **Results:** The study identified 55 shoulders with PJI, resulting in a 2.35% infection rate after primary shoulder arthroplasty in the young population. Male gender and reverse shoulder arthroplasty were risk factors for infection, while previous surgeries did not significantly contribute. The most common causative agents were *Cutibacterium acnes* and *Staphylococcus epidermidis*. Open washout had a 52.9% success rate for acute infections, while one-stage and two-stage revisions achieved infection control rates of 91.3% and 85.7%, respectively. Resection arthroplasty had an 81.8% success rate but poorer functional outcomes. **Conclusions:** PJI following shoulder arthroplasty in young patients is a significant concern. Preoperative planning should be carefully considered to minimize infection risk. Treatment options such as open washout and one-stage and two-stage revisions offer effective infection control and improved functional outcomes. Resection arthroplasty should be reserved for complex cases where reimplantation is not a viable option.

## 1. Introduction

Shoulder arthroplasty is the reference treatment for most degenerative conditions of the shoulder, with excellent reported results at mid- to long-term follow-up [1,2,3]. Recently, improvements in surgical technique and implant designs have made these procedures more and more reliable [4]. Shoulder arthroplasty has evolved significantly to address a variety of debilitating conditions, including osteoarthritis, rheumatoid arthritis, and complex fractures. It not only alleviates pain but also restores function and enhances the overall quality of life for patients [2,3].

Recent studies have shown a significant rise in the incidence of shoulder arthroplasty surgery. In France, 17,043 primary shoulder arthroplasty procedures were performed in 2018, marking a 47% increase from 2012 to 2018. Projections indicate a substantial rise in total shoulder arthroplasty procedures, which are expected to increase by 31% to 322% between 2018 and 2050 [5]. Moreover, the number of revision arthroplasty procedures reached 1508, reflecting a 39% increase from 2012 to 2018. Over the past decade in the United States, national hospital data reveal a notable surge in primary shoulder arthroplasty procedures, from 22,835 in 2011 to 62,705 in 2017, indicating a nearly three-fold increase over a six-year period. This pattern can be explained by various factors, including an aging population, heightened recognition of surgical advancements, and implant design innovations. The greatest increase in incidence was found in male patients and patients aged between 50 and 64 years [6].

In the last decade, a large body of evidence demonstrated that shoulder arthroplasty is a reliable and safe treatment in patients younger than 60 years, with substantial significant functional improvements and long implant survivorship [7,8]. As a result, the decision to operate on younger patients might be less subject to hesitation. Nevertheless, when performing shoulder arthroplasty in a young patient, much attention should be paid to the survivorship of the implants and complication occurrence over time. Among the complications after shoulder arthroplasty, infection is one of the more frequent, and probably the most devastating; it can lead to poor outcomes, prolonged hospital stays, and a considerable burden on healthcare systems [9,10,11].

Shoulder periprosthetic infections pose a significant challenge to both patients and healthcare providers. They have an incidence of 1–4% in primary arthroplasty and have a higher incidence after revision surgery [12,13,14]. Predisposing factors associated with periprosthetic shoulder infections include male gender, smoker, prior non-arthroplasty surgery on the affected shoulder, revision arthroplasty, shoulder arthroplasty due to traumatic situations, reverse shoulder arthroplasty, individuals receiving systemic or locally injected steroids, chemotherapy, rheumatoid arthritis, pathologic weight loss, obesity, peripheral vascular disease, and diabetes mellitus [9,13,15,16,17,18].

The most common bacterial microorganisms associated with shoulder periprosthetic infections are pathogens typically found in the normal skin flora of the axillary region, such as *Staphylococcus* species and *Cutibacterium acnes*. *Cutibacterium acnes* alone accounts for nearly 31% to 70% of all periprosthetic shoulder infections [15,19].

Treatment options for shoulder periprosthetic infection include debridement, antibiotic, and implant retention (DAIR), one-stage exchange, two-stage reimplantation, suppressive antibiotic therapy, and resection arthroplasty. The choice of treatment depends on multiple factors such as the timing of the infection, type of microorganism, soft-tissue and bone quality, patient’s age, co-morbidities, and expectations [9,20].

Despite numerous studies on periprosthetic infection, to the best of our knowledge, there has been no previous study specifically investigating periprosthetic infection in younger patients and evaluating functional outcomes. This study, based on a multicentric database of 1404 shoulders, aimed to describe the epidemiology and risk factors of shoulder periprosthetic infection in patients who have had a primary shoulder prosthesis before the age of 60, to analyze the treatment options and to evaluate the outcomes after treatment at a minimum 1-year clinical and radiographic follow-up.

## 2. Materials and Methods

### 2.1. Ethics

The study was approved by our institutional review board (IRB: 13B-T-SHOULDER-RM).

### 2.2. Patients

This was a multicenter retrospective and observational study that included 1404 shoulders with a primary arthroplasty performed on patients 60 years and younger (ranging from 18 to 60 years) from January 1991 to January 2016. For the present study, we identified, in the global population, all the shoulders that ever presented with a septic complication after the primary prosthesis or the subsequent revision surgeries. The diagnosis of infection was made according to the Musculoskeletal Infection Society criteria (identified pathologic agent, presence of a fistula, or combined biological markers) [21].

The exclusion criteria were as follows:Benign wound problems with spontaneous favorable outcomes.Aseptic revision surgery, with positive systemic bacteriological samples, but no postoperative septic complication (without any specific treatment).Primary prostheses for septic arthritis sequelae with positive intraoperative bacteriological samples, treated by adapted antibiotic therapy postoperatively, without any subsequent septic complication.

This work was divided into 2 studies: an epidemiologic study and an outcome study:

**For the epidemiologic study (Figure 1)**, we considered every patient of the global series with a history of periprosthetic joint infection, according to the previously described criteria. We identified 55 shoulders meeting these criteria. We collected preoperative information about the patients (age, sex, BMI, ASA score, significant medical history), and we particularly focused on surgical history (previous non-prosthetic surgeries, etiology, type of primary prosthesis, and revision procedures), as well as the bacteria involved and the time of onset of infection.

**For the outcome study (Figure 1)**, we only considered the patients with a minimum of 1-year follow-up after the last surgery performed for the treatment of infection. We identified 44 shoulders (11 shoulders having less than 1-year follow-up). In this population, we focused on the different treatment options (nonoperative treatment, open washout, 1- or 2-stage revisions, definitive explantation) and their outcomes. At the final follow-up, every patient had a clinical (pain, range of motion, Constant Score) and radiographic evaluation (status of humeral and glenoid implants). For each case, we determined if the infection could be considered healed or not based on clinical and radiographic data.

### 2.3. Statistical Analysis

Data were collected in EasyMedStat online database (version 3.24). Statistical analysis was performed with a 95% confidence level. We used a *t*-test to compare quantitative data, a chi-square test for qualitative data, or a Fisher test when groups were too small (n < 5). A *p*-value < 0.05 was considered significant.

## 3. Results

### 3.1. Epidemiologic Study

Among the 1404 shoulders included in the multicentric study, 55 shoulders had a reported history of periprosthetic infection after the primary procedure or after a subsequent revision surgery. That meant that when a shoulder prosthesis was implanted before the age of 60, the risk of infection during the surgical history of the shoulder at a mean follow-up of 95.8 months was 3.9%. Moreover, 33 cases of infection occurred after a primary arthroplasty, so the infection rate after a primary prosthesis in this young population was 2.35%.

We focused on the demographic data (age, sex, ASA score, BMI, medical and surgical history, follow-up) and tried to identify risk factors for infection (Table 1), comparing two populations: the infected shoulders (N = 55) and the non-infected shoulders (N = 1349). The age at the time of primary implantation was not associated with a higher risk of infection; in addition, BMI, ASA score, and medical history showed no association with a higher risk of infection. The main risk factor was the male gender (*p* = 0.001). Previous surgery, before the primary prosthesis, was not identified as a risk factor (*p* = 0.26). The distribution of the etiologies was not very different between the two populations (Table 2), except for aseptic osteonecrosis, which was clearly the etiology at lower risk of infection (no occurrence). There was a tendency for fracture sequelae to have a higher risk for infection, but it was not significant (*p* = 0.12). We also analyzed the surgical data (Table 3).

Among 55 infected shoulders, 4 had an infection recurrence after the first attempt of treatment by implant revision, so a total of 59 arthroplasties presented a postoperative infection. Among the 1662 reported procedures in the multicentric study, 1585 were prostheses that did not become infected. Revision prostheses represented 45.8% of the infected prosthesis population and only 18.4% of the non-infected prosthesis population (*p* < 0.001), meaning that revision surgery is at higher risk of postoperative infection. There was no significant influence of the approach, but this analysis must be considered with caution since the vast majority of the procedures were carried out through a deltopectoral approach (90%). Regarding the type of prosthesis, hemiarthroplasty (HA) was at lower risk of infection than total shoulder arthroplasty (TSA) and reversed shoulder arthroplasty (RSA). RSA was the implant at higher risk of postoperative infection (*p* < 0.001), but this should also be considered with caution since RSAs were revision prostheses in 47.2% of the cases (vs. 9.8% and 12.7% for HA and TSA, respectively).

Infection occurred after a mean delay of 41 months (range 0–236) after the implantation of the involved prosthesis. Sixteen infections occurred before 6 months (29.1%), and 10 within the 2 first months postoperative (18.2%). Because of the retrospective design of the study, we could not precisely report the clinical presentation at the time of diagnosis for each patient. The mean Constant Score (CS) was 28 ± 14 (range 10–66). Among resurfaced glenoids (TSA and RSA), 17 were loosened at the time of diagnosis (40.5%). The humeral implant was loosened in four cases (7.4%). The prosthesis was dislocated in only one case (1.8%). Two patients had negative bacteriologic samples (3.6%), but in these cases, the diagnosis of infection was confirmed by clinical presentation (fistula, one case) or histologic analysis (one case). In the 53 other cases, the identified bacteriologic agents (Figure 2) were more frequently *Cutibacterium acnes* (67.3%) and *Staphylococcus epidermidis* (27.3%). *Staphylococcus aureus* was isolated in only eight cases (14.5%), and other bacteria were isolated in five cases (9.1%) *(Enterococcus faecalis, Staphylococcus hominis, Pseudomonas aeruginosa)*. *Cutibacterium acnes* and/or *Staphylococcus epidermidis* were identified in 80% of the cases. In nine shoulders (16.3%), more than one bacteria species was isolated (polymicrobial infection). *Cutibacterium acnes* was isolated in 30 of 43 males (69.8%) and in 7 of 12 females (58.3%, *p* = 0.49).

### 3.2. Outcome Study

Among the 55 infected shoulders identified in the multicentric study, 44 had a minimum follow-up of 1 year after the last procedure. These 44 patients were included in the outcome study (Table 4). Two shoulders only had a medical treatment (no reoperation) and have been analyzed separately. We focused on the 42 shoulders that were reoperated on, at least one time, for the treatment of infection. We had a total of 51 procedures for analysis (Table 4).

Fifteen patients had a total of 17 open washouts (2 patients had 2 procedures) with or without component exchange, 23 patients had one-stage (9 cases) or two-stage revision (14 cases), and 11 patients had a definitive explantation. Six patients had more than one procedure (three patients had two procedures, and three patients had three procedures). After the surgery, all the patients had an adapted bi-antibiotic therapy for a minimum duration of 6 weeks. At a mean follow-up of 76 months (range 12–213), infection was considered healed in 38 cases (90.5%). The success rate of the 51 performed procedures was 76.5%.

The mean final CS was 42 ± 20 (range 10–85) and was significantly improved (gain of 14 points, *p* = 0.0005). The mean pain score was 10.7 ± 3.6 (range 5–15), and 17 patients (40.5%) had no or mild pain (score ≥ 13). The Constant Score was significantly better if the prosthesis remained in place (CS = 48) than if it was removed (CS = 27, *p* < 0.001). The functional outcomes were also better when the infection was eradicated (CS = 44) than when the shoulder remained infected (CS = 25, *p* = 0.02). There were nine reported complications (18%): six humeral fractures, two radial nerve palsies, and one case of instability. All these complications occurred after a revision surgery or a definitive explantation. At the final follow-up, 12% of the glenoid components (3/25) and 6.7% of the humeral stems (2/30) were loosened.

Fifteen patients underwent a total of 17 open washouts (14 as a first-line treatment, 1 after a first failed attempt of open washout, and 2 after a two-step revision procedure). The mean delay after index prosthesis was 19 months (0–239). The washed-out infected prosthesis was an HA in five cases, a TSA in three cases, and an RSA in nine cases. Removable components (humeral head in TSA, PE liner ± glenosphere in RSA) were changed in seven cases (41.2%). Among the nine RSAs washed out, one had both PE liner and glenosphere changed. Washout enabled infection control in nine cases (52.9%), including four cases with removable component exchange (44.4%), but in our series, component exchange did not significantly influence infection healing (*p* = 1). There was no significant influence of the type of prosthesis washed out (HA, TSA, or RSA). In cases of infection healing, washout had been performed at a mean of 2.8 months after index prosthesis, compared to a 37.1-month delay in cases of failure (*p* = 0.22). There were no intraoperative complications reported. Constant Score at the final follow-up has only been analyzed for the shoulders that healed after the washout procedure, without any further revision. For these eight shoulders, the mean CS was 57 ± 13 (range 32–78). The final functional outcomes of the seven patients who underwent further prosthesis revision were not representative of washout procedure outcomes and thus were not analyzed. When subsequent revision was performed after a failed washout, infection healing ultimately occurred in four out of six cases (66.7% vs. 89.3% for first-line revision surgery, *p* = 0.20).

Resection arthroplasty was performed in 11 cases, with (4) or without (7) interposition of a cement spacer. In seven cases, it was a first-line treatment; in three cases, it followed open washout failure; and in one case, it was performed after failure of a two-stage revision. The explantation was performed after a mean delay of 57 months (6–241). The explanted prosthesis was an HA in two cases, a TSA in three cases, and an RSA in six cases. The average CS was 27 ± 14 (range 10–51). There was no significant difference between the shoulders in which a spacer was implanted and those that were simply explanted (CS = 25 vs. 28, *p* = 0.79), but among the four spacers, one was broken, and one was found to have migrated at the final follow-up, with a mean CS of 15 (range 10–20) for these two cases. The complication rate was 27.3% (two intraoperative fractures and one definitive and total radial nerve palsy). Nine shoulders were considered healed at the final follow-up (81.8%).

One-stage revision was performed in nine cases, always as a first-line treatment, after a mean delay of 22 months (range 6–84). The reimplanted prosthesis was an HA in one case, a TSA in three cases, and an RSA in five cases. All the patients were considered healed at the final follow-up (100%). The complication rate was 11.1% (one radial nerve palsy, after RSA implantation). The average final Constant Score was 45 ± 24 (range 10–85).

Two-stage revision was performed in 14 cases, as a first-line treatment in 12 cases and after a failed attempt of open washout in 2 cases. The mean delay for revision was 40 months (range 2–154). The prosthesis reimplanted at the second stage was an HA in five cases (three metal anatomic HAs, one snooker ball, and one hemi-RSA) and an RSA in nine cases. This procedure allowed infection control in 12 of the 14 cases (85.7%). The complication rate was 35.7% (four intraoperative humeral fractures, one instability). The average final Constant Score was 45 ± 17 (range 20–73), but one shoulder was not included in this CS analysis because it was reoperated on for definitive explantation and had no implant at the final follow-up. There was no significant difference between one-stage and two-stage revisions regarding healing rate (*p* = 0.50), final CS (*p* = 0.97), delay before revision (*p* = 0.24), or complication rate (*p* = 0.34), even if two-stage revisions seemed to have a slightly higher complication rate. Further analysis of outcomes according to the type of prosthesis reimplanted was not possible because of the small number of cases in each of the subgroups. TSA was never reimplanted after a two-stage revision. Finally, two shoulders (one HA and one TSA) were not reoperated on because of poor medical condition, low impairment, and/or patient refusal.

At a mean follow-up (after diagnosis of infection) of 351 months (range 267–435), the average Constant Score was 32.5 (32–33), which was lower than that after revision surgery (CS = 45, *p* < 0.05) but equivalent to (or slightly better than) functional results after resection arthroplasty (CS = 27, *p* = 0.21). There were no reported general complications due to chronic infection. At the final follow-up, the humeral stems were stable, but the glenoid implant of the TSA was loosened. Moreover, among the 42 reoperated patients, 4 did not heal and could be evaluated after nonoperative treatment at 78 months (17–156) follow-up after the last procedure (two explantations, one TSA, and one RSA). The average Constant Score was 25 (range 15–37) at the final follow-up, and no general complications were reported due to chronic infection.

## 4. Discussion

### 4.1. Epidemiology

This study reported a 2.35% rate of infection after primary shoulder arthroplasty in patients under the age of 60. The rate was 3.9% if we consider the risk of having a periprosthetic infection during the course of shoulder history (including eventual revision procedures), which is concerning, and higher than reported rates after hip or knee arthroplasty, and even after TSA [9,14,22,23,24,25]. Richards et al. previously reported a 5% higher risk of infection after shoulder arthroplasty with every 1-year decrease in age [15].

Knowing how devastating the consequences of shoulder periprosthetic infection can be, we should be aware of the potential risk factors and of the ways to avoid, or minimize, them. Some of the risk factors we identified cannot be controlled, such as the male gender and the length of follow-up. The increased risk in male patients has already been reported [15,16,17,20]. These factors can only make us more vigilant when making the decision to operate on a young patient, especially if it is a man, because the history of the shoulder extends for many years after the primary prosthesis, with potential revision surgeries, and one day, a periprosthetic infection. Curiously, surgical history before the first arthroplasty was not identified as a significant risk factor in our study, even if it is known to be in the literature [18]. The etiology had a low influence on the risk of infection, but we could suspect that the risk is a little bit higher with fracture sequelae, even if not significant (surgical history, complex bone deformity, soft-tissue alteration), like it was previously reported by Richards et al. [15]. It seemed on the contrary that aseptic osteonecrosis was at very low risk. The type of implant also influenced the risk of postoperative infection. HA was significantly at lower risk, whereas RSA was more at risk of postoperative infection. The infection rate after RSA has often been reported to be higher than that for other shoulder prostheses [20,26,27,28,29,30,31,32,33], probably because of the characteristics of the implants, the subacromial dead space, and the frequent complexity of the cases requiring an RSA. Indeed, 45% of the implanted RSAs in our series were revision prostheses. Nevertheless, and as already reported in numerous studies [20,30,34], revision surgery, which accounted for nearly 50% of the infection cases, was a significant risk factor for periprosthetic infection. All this should make us think twice when choosing the type of implanted prosthesis and before scheduling a revision surgery.

As reported in most of the publications about postoperative shoulder infection [20,23,32,35,36,37,38,39,40,41,42,43,44,45,46,47,48,49,50], we identified a majority of *Cutibacterium acnes* and *Staphylococcus epidermidis* (80%), but we did not find a higher prevalence of *C. acnes* in the male patients. The high prevalence of *P. acnes* and other slow-growing bacteria is responsible for low-grade infections and atypical clinical presentations, so the diagnosis of infection is not always obvious [51]. We should always think that a revision for an aseptic complication might be an occult infection. Therefore, we can imagine that the reported infection rate in our study may be underestimated. Moreover, the specificity of the bacteria involved in shoulder periprosthetic infection should lead us to apply adapted prevention measures (chlorhexidine skin preparation, reoperative application of benzoyl peroxide, intra-articular injection of antibiotics, etc.) to limit the skin and sebaceous gland colonization and thus limit the inoculum into the joint during the surgery.

### 4.2. Management and Outcomes

Among the 44 infected shoulders included in the outcome study (>1-year follow-up), only 86.4% were considered healed at the last follow-up, after at least one line but sometimes two or three lines of surgical treatment.

The functional outcomes were far lower than those reported after shoulder arthroplasty, and the complication rate in reoperated patients was high (18%). This highlights the fact that infection is a severe complication, which is difficult to treat, and often leads to functional deterioration. To elaborate the best strategy for each patient, surgeons should be aware of the different treatment options and their consequences. Years ago, definitive explantation was often the preferred option after a shoulder periprosthetic infection [29,32]. In the last ten years, several authors advocated implant revision rather than resection arthroplasty and demonstrated this was technically possible, even if more challenging [20,26,37,38,39,47,48,49,50]. Our results also argue for implant preservation or replacement, rather than removal, in order to preserve shoulder function.

Washout was the least effective option, allowing infection control in only 52.9% of the cases. Low rates of success after washout have been previously reported [26,47], especially for late and chronic infections. However, infection healing after a simple washout procedure leads to the best functional result at the final follow-up. Moreover, there was no reported complication, and in case of failure, open washout did not compromise the success of subsequent procedures. In a previous study, it was suggested that the washout success rate in RSA could be increased by the exchange of all components that can be easily removed, such as the polyethylene liner and glenosphere [20]. In this series, the humeral head (in TSA) or polyethylene liner (in RSA) was changed in only seven cases (41.2%), and both the liner and glenosphere were changed in only one case, suggesting that this 52.9% rate does not reflect what we could obtain with “optimized washout”. The time of onset of the infection also influenced the results of open washout on infection healing, making it a good option for acute infections, as a first-line treatment. Nevertheless, Jacquot et al. [20] reported no influence of the delay on the efficiency of open washout, in a series of infected RSAs, and recommended “optimized open washout” as a first-line treatment for every case, whatever the delay. In our opinion, washout remains an acceptable first-line treatment for infection, but preferably when seen early (<3 weeks), providing that implants are not loosened and that debridement is extensive, with the exchange of all easily removable components, such as humeral head, PE liner, glenosphere, and eventually the humeral stem, which can be easily done with the new short and uncemented stems. And then, open washout could be not so far from one-stage revision, thanks to the evolution of the implant design.

Revision seemed to be the best therapeutic option, even if technically challenging and at higher risk for complications. The functional outcomes were satisfying (mean CS = 45), without a compromise in infection control (91.3%). Recent publications also advocate the fact that revision should always be considered in the face of an infected prosthesis, rather than resection arthroplasty [26,37,38,45,47,49]. In our series, one- and two-stage revisions had comparable results. Nevertheless, the complication rate after one-stage procedures seemed slightly lower, even if not significantly. Previously, Jacquot et al. [20] found similar results in a series of infected RSAs, and Beekman et al. [38] reported very promising results with one-stage procedures. As it is demonstrated to be efficient and at lower surgical risk, one-stage procedures may be considered if the bacteriologic agent is known preoperatively and there is no major bone or soft-tissue defect. For the other and more complex cases, two-stage revision remains indicated.

The type of reimplanted prosthesis should be decided according to the type of explanted prosthesis, soft-tissue conditions, glenoid and humeral bone stock, and patient functional expectations, knowing that a TSA can rarely be reimplanted after a two-stage procedure. Resection arthroplasty was probably the worst option, leading to a functional disaster, and infection healing was not always obtained (81.8%). This was in accordance with previous publications reporting very low outcomes but quite efficient control of infection [29,39,43,52,53]. The functional result was not influenced by the use of a spacer, as shown by Verhelst et al. [53], but the implantation of a spacer was vulnerable to long-term mechanical complications (migration, breakage) and was associated with lower functional outcomes. Definitive explantation should remain a salvage procedure, when revision is not possible (major bone defects, fragile patient) and when the infection affects the patient’s general condition, implying that nonoperative treatment is not a good option.

Finally, some patients [36] have been treated nonoperatively after the diagnosis of infection, and some others [38] were still infected after a revision surgery. The decision was made not to reoperate on these patients, because of poor medical condition, patient refusal, and/or preserved function. Outcome analysis of these patients, after quite a long follow-up, showed that function was not acceptable and seemed better for the non-reoperated patients than for those who remained infected after a failed surgical procedure. Moreover, none of these patients had any general medical complications related to chronic infection. Nonoperative treatment may be an acceptable therapeutic option and must be considered in some selected cases, especially when the function of the shoulder is acceptable.

This study has several limitations, notably a relatively small sample size of periprosthetic infection among 1404 operated shoulders. A formal statistical design power analysis to estimate the sample size for the relevant statistical analyses was not conducted. Consequently, there might be limitations in the ability to detect smaller, yet potentially meaningful, effects due to the sample size not being explicitly determined based on statistical power calculations. However, this can be forgiven due to the rarity of the indication and the complications specific to this group. Moreover, data accuracy, owing to the retrospective design, relied on medical records and was susceptible to human error. The multicenter nature of the study could affect treatment variability across cases, potentially influencing the final outcomes. Furthermore, the study primarily focused on infection rates, neglecting broader aspects such as patient-reported outcomes and quality of life, providing a narrower perspective on postsurgery experiences. Future research addressing these limitations is crucial for a comprehensive understanding of periprosthetic joint infections in young shoulder arthroplasty patients.

## 5. Conclusions

Infection after shoulder arthroplasty is a frequent, and devastating, complication. The risk increases with the length of the surgical history of the shoulder, which is a real concern in young patients. These findings emphasize the importance of meticulous preoperative planning, and great caution must be taken in deciding on shoulder replacement, as well as the type of prosthesis implanted to mitigate infection risks. The risk of PJI after revision is common, and surgeons should be aware of the possibility of future revisions in this young population, which may increase the risk of future PJI.

This comprehensive analysis contributes valuable insights to the medical community, guiding clinicians in optimizing patient outcomes and minimizing the impact of PJI in the context of shoulder arthroplasty for younger patients. Moreover, our study advocates for the careful selection of appropriate treatment strategies. When the infection occurs, surgical decisions should be based on several parameters (type of prosthesis, bone or soft-tissue defect, time of onset of infection, patient general condition). Open washout is a receivable option as a first-line treatment, providing that the prosthesis is not loosened and that the surgeon proceeds to exchange all the removable components. One- or two-stage revision is the more efficient surgical procedure but is technically more challenging and may sometimes lead to severe complications. Definitive explantation should remain exceptional, for complex and desperate cases only, especially in young patients since it always leads to functional disaster.

## Figures and Tables

**Figure 1 microorganisms-11-02770-f001:**
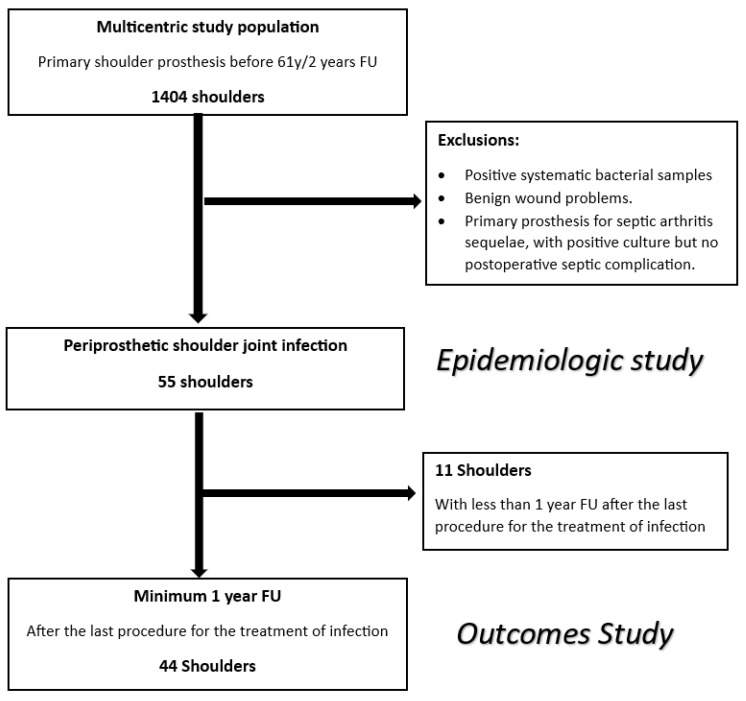
Flowchart.

**Figure 2 microorganisms-11-02770-f002:**
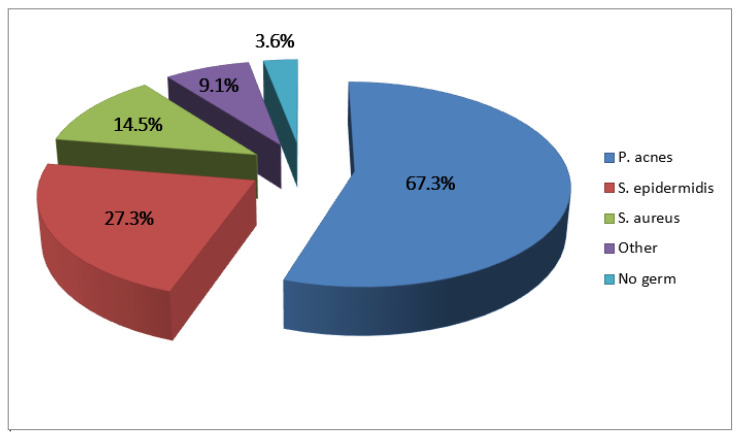
Identified bacteria.

**Table 1 microorganisms-11-02770-t001:** Demographic data and risk factors.

	Infected Shoulders (N = 55)	Non-Infected Shoulders (N = 1349)	*p*
Mean FU (months) (since 1st prosthesis)	157	93	
Mean age at inclusion	49.8	50.8	>0.05
Sex:			
Male	78.2%	51.6%	0.001
Female	21.8%	48.4%	
BMI	27.5	27.5	>0.05
ASA score			
ASA 1	27.5%	36.4%	
ASA 2	60%	51.2%	>0.05
ASA 3	12.5%	12.2%	
ASA 4	0%	0.2%	
Medical history			
Diabetes	11.6%	6.6%	0.21
Smoker	29.3%	25.6%	0.59
Surgical history (before 1st prosthesis)	42.6%	35.1%	0.26

FU: follow-up, BMI: body mass index.

**Table 2 microorganisms-11-02770-t002:** Primary diagnosis distribution and infection rate.

	Infected Shoulders (N = 55)	Non-Infected Shoulders (N = 1349)	*p*	Infection Rate in Shoulder History	Infection Rate after Primary Prosthesis
Fracture sequelae	16 (29.1%)	276 (20.4%)	0.12	5.5%	2.7%
PGHOA	15 (27.3%)	368 (27.3%)	0.99	3.9%	2.3%
Instability arthropathy	6 (10.9%)	159 (11,8%)	0.84	3.6%	1.8%
Acute fracture	6 (10.9%)	124 (9.2%)	0.66	4.6%	3.1%
Rheumatoid arthritis	5 (9.1%)	125 (9.3%)	0.96	3.8%	3.1%
MCT/CTA	3 (5.5%)	84 (6.2%)	1	3.6%	2.3%
Tumor	1 (1.7%)	34 (2.5%)	1	2.9%	2.9%
Aseptic osteonecrosis	0 (0%)	132 (9.8%)	0.007	0%	0%
Other diagnosis	3 (5.5%)	47 (3.5%)	-	-	-

PGHOA: primary glenohumeral arthritis. MCT: massive cuff tear (Hamada 1, 2 3). CTA: cuff tear arthropathy (Hamada 4, 5).

**Table 3 microorganisms-11-02770-t003:** Surgical data. Population No. 1 (N = 59): Among the 55 infected shoulders, 4 had a recurrence of infection after a revision surgery (new prosthesis), so 2 arthroplasties were followed by postoperative infection in the same shoulder history. Consequently, we analyzed 59 postoperative infections in 55 shoulders. Population No. 2 (N = 1585): This population is the global series of 1662 reported operations, from which we removed the infected arthroplasties (population No. 1, N = 59), and the definitive explantations (no implant).

Total (N = 1662)	Arthroplasty with Postoperative Infection (N = 59)	Arthroplasty with No Postoperative Infection (N = 1585)	*p*
Primary/Revision	32 (54.2%)/27 (45.8%)	1294 (81.6%)/291 (18.4%)	<0.001
Approach			1
- DP		1444 (91%)	
- SUP	53 (96.4%)	54 (3.4%)	
- Other/NC	2 (3.6%)	88 (5.6%)	
Type of Arthroplasty:			
HA	14 (23.7%)	679 (42.8%)	0.003
- Primary	13 (92.8%)	612 (90.1%)	
- Revision	1 (7.2%)	67 (9.9%)	
TSA	19 (32.2%)	557 (35.1%)	0.24
- Primary	13 (68.4%)	490 (88%)	
- Revision	6 (31.6%)	67 (12%)	
RSA	26 (44.1%)	349 (22.1%)	<0.001
- Primary	6 (23.1%)	192 (55%)	
- Revision	20 (76.9%)	157 (45%)	

HA: hemiarthroplasty. TSA: total shoulder arthroplasty (unconstrained, anatomic). RSA: reversed shoulder arthroplasty. DP: deltopectoral approach. SUP: superior approach.

**Table 4 microorganisms-11-02770-t004:** Management and outcomes.

Procedure	N	Infection Cured (%)	Last-FU	Prosthesis Reimplanted	Complication (%)
**Open Washout**	17	52.9	57	-	0
**Resection Arthroplasty**	11	81.8	27	4 spacers	23.5
**1-Stage Revision**	9	100	45	1 HA, 3 TSA, 5 RSA	11.1
**2-Stage Revision**	14	85.7	45	5 HA, 9 RSA	35.7
**1- and 2-Stage Revisions**	23	91.3	45	6 HA, 3 TSA, 14 RSA	26.1
**Nonoperated**	2	-	33	-	0
**Comparisons of “Last-FU” between Groups**					
**Resection Arthroplasty vs. 1-Stage Revision**			27 vs. 45		*p* = 0.06
**Resection Arthroplasty vs. 2-Stage Revision**			**27 vs. 45**		***p* = 0.01**
**Resection Arthroplasty vs. 1- and 2-Stage Revision**			**27 vs. 45**		***p* = 0.01**
**1-Stage Revision vs. 2-stage Revision**			45 vs. 45		*p* = 0.97

FU: follow-up. HA: hemiarthroplasty. TSA: total anatomic shoulder arthroplasty. RSA: reversed shoulder arthroplasty.

## Data Availability

No new data were created or analyzed in this study. Data sharing is not applicable to this article.

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
