# Peer review of "Infected Shoulder Arthroplasty in Patients Younger than 60 Years: Results of a Multicenter Study"

_microorganisms, 2023, doi:10.3390/microorganisms11112770_

Round 1
Reviewer 1 Report
Comments and Suggestions for Authors
Jacquot et al Microorganisms -MDPI- 091923
Jacquot et al evaluated periprosthetic joint infection following primary shoulder arthroplasty retrospectively from 1404 shoulders in patients under 60 at least 1-year follow-up after infection treatment. They reported ~2.4 % infection rate with higher risk factors in males, longer follow up, and reverse shoulder arthroplasty but not previous surgeries. They identified Cutibacterium acnes and Staphylococcus epidermidis as the most common causative agents. They concluded that preoperative planning and treatment using open washout in one-stage and two-stage revisions are beneficial to control infection and improve functional outcomes. They claimed that resection arthroplasty should be reserved for complex cases. Comments are listed below.
(1) Abstract: Clear summary of the study but the conclusion “resection arthroplasty should be reserved for complex cases” seems ambiguous- what do they mean by complex cases?
(2) Introduction:
a) Background information is sparse and does not include state of the art, consensus background information on shoulder arthroplasty, treatments and complications.
b) It would be helpful if the literature information on the potential sources of infection and what microorganisms have been identified in situ was included.
c) Their aim was to describe epidemiology and risk factors of shoulder periprosthetic infection in patients (< 60) who had a primary shoulder prosthesis, analyze the treatment options and outcomes. However, would this aim be achieved with the number of patients studied?
(3) Method
a) What was the statistical design power estimates of sample size for relevant statistical analyses?
b) What was the age range of patients included. Wide range could contribute to heterogeneity in the study group. Also, what was the time included before 2016?
c) The multi-center study yielded 55 infected shoulders from 1404 shoulders for analysis. Would this number be sufficient for an “epidemiologic and outcome” study?
d) Are comorbidities not considered in inclusion/exclusion criteria?
(4) Results
a) Epidemiologic data collected and outcomes appear straightforward. Since 55 shoulders were identified as infected, would age breakdown be informative?
b) Would the level of patient activity, possible trauma could predispose to infection? It would be interesting to address these possibilities in the discussion.
(5) Discussion -conclusion
a) The novelty added value of their study was not adequately addressed.
b) The limitations of their study were not appropriately discussed.
c) Overall, the present paper offers interesting information on shoulder arthroplasty in a younger patient group.
Reviewer 2 Report
Comments and Suggestions for Authors
The text is overall well written although too length: some periods could be shortened and the text made lighter for an easier read, especially in the results part.
line 40: change "Improvements" to "improvements"
lines 92-93: you assume the sample has a normal distribution: did you perform test for normality before using the T test?
lines 92-93: which quantitative data did you mean? If within the same subject, a paired t test would have been used; in case of independent samples, a T test for non paired sample would have been used: could you specify in the text please?
lines 101-103: primary and revision procedures should not be considered together since it is well known that revision procedure carry out a higher risk for infection due to both previous surgery and longer operative time. You should separate this sample in two distinct subgroups.
lines 111-112: how could longer follow-up times be considered as risk factors for infection? This is meaningless, since it is logical that subclinical forms of infection may become more evident over time.
line 142:Staphylocoque : please change to staphylococcus
lines 331-332: please specify "early" in days. Usually debridement, lavage and PE exchange ins indicated for infections occurring within the 15 days from the surgical procedure
Two stage revision includes the use of antibiotic loaded spacer: did you use homemade spacers? how long were the spacers kept in situ? How did you consider time for revision after spacer? Such data should be reported since inaccurate timing or pre operative planning may increase the risk for infection
Comments on the Quality of English Languageminor changes in the highlighted parts
Reviewer 3 Report
Comments and Suggestions for Authors
This paper reports data about the epidemiology and the management of the infected shoulder arthroplasty in patients younger than 60 years. These data are useful, and they agree with the current literature. However, the article is not easy to follow and I wonder whether the statistical methods used are optimal. In more detail:
· I believe the title should be “Results of a Multicenter Study” instead of “Results of the Multicenter Study”.
· The text is really difficult to read, it should be separated in paragraphs.
· The introduction section should be larger, at least 3 paragraphs.
· In my point of view, a multilogistic statistical analysis should be performed, to eliminate confounding between the independent variables.
Round 2
Reviewer 1 Report
Comments and Suggestions for Authors
The authors responded to my critique adequately, and revised their manuscript accordingly. I recommended acceptance of the revised manuscript as is.
Author Response
Dear reviewer,
We would like to sincerely thank you for your valuable time and expertise in re-revising our manuscript, which significantly improved the quality of our study. After modifying the first draft, we totally agree with all the suggested comments, which were vital for our manuscript.
Thank you again.
Best regards,
Reviewer 3 Report
Comments and Suggestions for Authors
The authors have addressed most of the comments. The manuscript is clearly improved and now it is more readable. However, the last comment about the statistical analysis has been insufficiently answered. A multivariable analysis is invaluable for eliminating confounding factors and present more robust results. In my opinion the authors should recruit a statistician to conduct the analysis, having no expertise is an inadequate answer. I would be more than happy to promote the manuscript for publication once the statistical analysis is improved.
Author Response
Dear reviewer,
Thank you for your feedback and suggestion regarding the use of multilogistic statistical analysis to eliminate confounding between the independent variables. We appreciate your insightful input into our research.
Our apologies for the answer we gave about our inability to manage the multilogistic statistical analysis. This latter was not clear. We considered carefully your comment with the biostatician of our team, but especially because of the heterogeneous etiologies, as well as the small numbers of patients in each group, we were unable to perform the suggested multilogistic analysis and were not convinced that it could lead to better informations for our medical community. That's the reason why we talked about maybe a lack of expertise for treating this very specific stastitical analysis for small samples of patients.
Moreover, we would like to highlight that there was no statistically significant difference between the groups in Table 1, except for males, which were more common in the infected shoulder groups. This finding is consistent with previous publications. Additionally, the distribution of etiologies did not show a statistically significant difference in infection rates, except for aseptic osteonecrosis, which was lower and has been addressed in our discussion. It is important to note once again that the sample size for the infection group remains small, and performing multivariate analysis would not yield meaningful changes to the existing results.
Regarding Table 3, the higher infection rate in RSA cases was extensively discussed, emphasizing the higher number of revision cases in RSA (lines 350-354). As for Table 4, the results clearly indicate that resection arthroplasty has a poorer outcome compared to revision, and this difference is statistically significant, which aligns with expectations.
Furthermore, we acknowledge the limitations of our study, including the issues related to controlling confounding variables. However, given the context and the scope of our research, it seems not contributive to perform multilogistic analysis for this study.
Thank you once again for your valuable feedback and understanding.
Sincerely